# Forecasting COVID-19 Severity by Intelligent Optical Fingerprinting of Blood Samples

**DOI:** 10.3390/diagnostics11081309

**Published:** 2021-07-21

**Authors:** Simão P. Faria, Cristiana Carpinteiro, Vanessa Pinto, Sandra M. Rodrigues, José Alves, Filipe Marques, Marta Lourenço, Paulo H. Santos, Angélica Ramos, Maria J. Cardoso, João T. Guimarães, Sara Rocha, Paula Sampaio, David A. Clifton, Mehak Mumtaz, Joana S. Paiva

**Affiliations:** 1iLoF—Intelligent Lab on Fiber, Limited, Oxford OX1 2EW, UK; sfaria@ilof.tech (S.P.F.); ccarpinteiro@ilof.tech (C.C.); vpinto@ilof.tech (V.P.); srodrigues@ilof.tech (S.M.R.); jalves@ilof.tech (J.A.); fmarques@ilof.tech (F.M.); mlourenco@ilof.tech (M.L.); psantos@ilof.tech (P.H.S.); srocha@ilof.tech (S.R.); psampaio@ilof.tech (P.S.); mmumtaz@ilof.tech (M.M.); 2Departamento de Bioquímica, Faculdade de Medicina da Universidade do Porto, 4200-319 Porto, Portugal; jtguimar@med.up.pt; 3Serviço de Patologia Clínica, Centro Hospitalar Universitário de São João, 4200-319 Porto, Portugal; angelica.ramos@chsj.min-saude.pt (A.R.); u009550@chsj.min-saude.pt (M.J.C.); 4EPIUnit—Instituto de Saúde Pública da Universidade do Porto, 4050-600 Porto, Portugal; 5i3S—Instituto de Investigação e Inovação em Saúde, Universidade do Porto, 4200-135 Porto, Portugal; 6IBMC—Instituto de Biologia Molecular e Celular, Universidade do Porto, 4200-135 Porto, Portugal; 7Department of Engineering Science, Institute of Biomedical Engineering, University of Oxford, Oxford OX3 7DQ, UK; davidc@robots.ox.ac.uk; 8Instituto de Ciências Biomédicas Abel Salazar, University of Porto, 4200-319 Porto, Portugal

**Keywords:** optical fingerprinting, photonics, machine learning, predictive biomarker, COVID-19

## Abstract

Forecasting COVID-19 disease severity is key to supporting clinical decision making and assisting resource allocation, particularly in intensive care units (ICUs). Here, we investigated the utility of time- and frequency-related features of the backscattered signal of serum patient samples to predict COVID-19 disease severity immediately after diagnosis. ICU admission was the primary outcome used to define disease severity. We developed a stacking ensemble machine learning model including the backscattered signal features (optical fingerprint), patient comorbidities, and age (AUROC = 0.80), which significantly outperformed the predictive value of clinical and laboratory variables available at hospital admission (AUROC = 0.71). The information derived from patient optical fingerprints was not strongly correlated with any clinical/laboratory variable, suggesting that optical fingerprinting brings unique information for COVID-19 severity risk assessment. Optical fingerprinting is a label-free, real-time, and low-cost technology that can be easily integrated as a front-line tool to facilitate the triage and clinical management of COVID-19 patients.

## 1. Introduction

The coronavirus disease 2019 (COVID-19) pandemic has generated a surge in critically ill patients who progress rapidly to respiratory collapse, shock, and multiple organ dysfunction or failure [1,2,3]. Severe or critical illness develops approximately one week after the onset of symptoms in about 20% of the patients diagnosed with COVID-19 [4]. The risk for severe disease increases among elderly people, women, and people with underlying chronic health conditions, such as diabetes mellitus, immunosuppression, obesity, cardiovascular, or respiratory disease [5,6,7,8]. Yet, young healthy people may also become critically ill. Severe COVID-19 disease may be associated with clinical and biochemical signs of inflammation, namely high fever, thrombocytopenia, hyperferritinemia, and increased C-reactive protein and interleukin-6 levels [9,10]. However, there is no biomarker that can predict severe disease at the time of diagnosis.

Despite great efforts being made to understand the disease course and molecular mechanisms of disease, COVID-19 treatments are not efficient, and their effectiveness is known to vary according to disease severity [11,12,13,14]. The early identification of patients that are at risk of developing severe symptoms may enable adequate treatment strategies and targeted recruitment for randomized controlled trials, improving their success rate. Furthermore, the high number of COVID-19 severe cases has globally overwhelmed healthcare systems, leading to a shortage of trained professionals, equipment, and beds in intensive care units to admit all critical cases [15]. Screening of COVID-19 patients during their initial hospital visit for risk of severe disease is key to inform clinical decision making and assist resource allocation, mainly in critical care units.

Here, we demonstrate the clinical utility of a label-free, real-time, and low-cost technology that agnostically analyzes blood samples of COVID-19 patients at their first hospital presentation to forecast disease severity. This platform, termed intelligent Lab on Fiber (iLoF), analyzes serum specimens through a highly focused laser beam, guided by a micro-lensed optical fiber, to obtain its unique backscattered signal signature [16,17]. This signal captures the specific combination of the Brownian motion of particles present in the sample under the influence of a harmonic electromagnetic potential generated by the electromagnetic field propagated by the polymeric micro-lensed fiber tip [18]. The backscattered signal of the sample can then be analyzed to retrieve information on the biophysical properties of particles smaller than the wavelength of the irradiation light (976 nm), such as their microscopic refractive index or optical polarizability [18]. To extract the information that might predict COVID-19 disease severity, the collected digital signal was processed and 98 time- and frequency-domain features were calculated (patient optical fingerprint). Prediction of ICU admission was framed as being a binary classification problem using machine learning models. Baseline models using the optical fingerprint information (model 1) and patient comorbidities/age (model 2) were used to build a stacking ensemble model of COVID-19 severity risk assessment. This technology is currently being tested in clinical settings to support clinical management decisions and to predict patient inflow in advance.

## 2. Materials and Methods

### 2.1. Description of the Prospective Study

This study was prospectively conducted in compliance with the Helsinki declaration and following the relevant regulations regarding the use of human samples. The study was approved by the Ethics Committee of Centro Hospitalar Universitário São João (CHUSJ; Porto, Portugal) under the protocol number 131/2020, and all patients provided informed consent. Patients included in the study were admitted at CHUSJ between May and August 2020, with confirmed COVID-19 positive diagnosis by RT-PCR assay of SARS-CoV-2 for nasopharyngeal swab specimens. Peripheral blood samples, demographic information (age, gender), comorbidities, and laboratory data (lymphocyte and leukocyte counts, C-reactive protein level, IgG detection) were collected at the moment of diagnosis. Patients presented early/mild symptoms of disease at the moment of diagnosis, and the decision of hospital admission was made considering the demographic, clinical, and laboratorial patient analysis. Multiple sample collection was not required for this study. Patients were followed for up to 4 weeks from admission to determine the primary outcome (admission at ICU). During the follow-up period, 38 patients developed severe disease symptoms requiring admission at ICU; 25 patients presented mild symptoms and required hospitalization but not intensive care; 25 patients displayed only light symptoms and recovered at home. In total, 88 patients were enrolled in the study. The researchers who collected patient demographic, laboratory, and outcome data were not involved in the final data summary and analysis.

### 2.2. Sample Collection and Preparation

Whole blood samples collected in BD PPT™ Tubes or BD SST™ Tubes (for plasma and serum preparation, respectively; BD Biosciences, CA, USA) were immediately and gently inverted 5 times. To obtain serum, BD SST™ Tubes were maintained upright for at least 30 min. Thereafter, samples were centrifuged at 800× *g* for 10 min, at room temperature (RT). Undiluted plasma samples were transferred to a new tube to obtain serum preparations or stored at −20 °C until further use. To convert plasma samples into serum, undiluted plasma was treated with thrombin (5 U/mL, System Biosciences, CA, USA), for 5 min at RT, with gentle mixing. Samples were centrifuged at 9600× *g* for 5 min, and the supernatant transferred to a new clean tube. Serum samples were diluted 1:2 in 1× phosphate-buffered saline (PBS; Corning, NY, USA) solution and 300 µL of diluted sample were pipetted onto a µ-Dish 35 mm, low wall (ibidi GmbH, Gräfelfing, Germany) to be analyzed at the iLoF platform. No further sample preparation was required.

### 2.3. Probe Fabrication

The sensing probes used in this work were microlenses fabricated at the apex of optical fibers through a guided-wave photo-polymerization process [19]. These microlenses were produced in-house, in the optical setup depicted in Appendix A (Step 1). This comprises a 405 nm laser diode (LuxX CW, 405 nm, 60 mW, Omicron, Rodgau-Dudenhofen, Germany), whose beam is reflected at 90° by two mirrors (RC-10Q620B, Newport, CA, USA). The beam is then focused by a 10× objective (RC-M-10X, Newport, CA, USA) to allow coupling to a cleaved optical fiber (SM980-5.8-125; Thorlabs, NJ, USA). Since the fiber is multimode at 405 nm, the angle upon which the beam is incident in the fiber core will determine the propagation modes and, consequently, the pattern at the output of the fiber.

The fabrication of the probes comprised two main steps. Firstly, a cleaved optical fiber was dipped in a drop of 0.3% concentrated solution of photo-initiator (IRGACURE 819) diluted in pentaerythritol triacrylate monomer (PETIA), and placed on a flat surface. Both chemicals were kindly provided by Dr. Olivier Soppera from the Institut de Science des Matériaux de Mulhouse, Mulhouse, France. Then, the 405 nm laser light coupled to the optical fiber was turned on to expose the polymer drop, inducing the linking of the monomers. The drop was exposed for 10 s with an optical power of 4 µW. After the exposure, the unpolymerized solution was rinsed off with ethanol (96%), revealing a tall polymeric lensed tip. Due to the self-guiding effect of the process, the resulting structure had the same diameter as the core of the optical fiber, with a spherical shape at the top, imprinted by the fundamental optical mode (LP01) that propagated along the fiber. Given the high aspect ratio of this structure, a second was added to increase the contact surface between the polymer and the optical fiber, thus guaranteeing maximum resistance to successive dips in serum samples. Previous studies have shown that this structure did not affect the guiding properties of the central polymeric lensed structure [20]. This second structure was achieved by dipping the optical fiber a second time in a similar solution, this time 2% concentrated, while leaving the extremity of the initial tip outside the drop. External exposure of the drop was then conducted at 20 µW for 3 min. After rinsing it again with ethanol, the resulting structure was similar to a dome protecting the tall spherical lens (Appendix A).

### 2.4. Data Acquisition

All samples were analyzed in an in-house built optical setup (Appendix A). Briefly, the setup was composed of a pigtail 976 nm laser diode (S28-7602-500, Lumentum, CA, USA) connected to a 1:99 optical coupler (1×2 SM-coupler, 980nm, 01/99, HI1060 fiber, FC/APC, Laser Components, Olching, Germany). The 1% port of this coupler was connected to a photodetector (PDA36A2, Thorlabs, NJ, USA) to allow real-time laser signal monitoring. The 99% port guided the signal to a 50:50 optical coupler (FOSC-1-98-50-L-1-H64F-2, AFW Technologies, Victoria, Australia). This coupler was also connected to the polymeric microlens and to a photodetector, to allow for the simultaneous sample illumination and collection of the backscattered radiation. Additionally, the system encompassed an inverted optical microscope to monitor both the sample under analysis and the probe integrity. Shortly, it was composed of a 20× objective (RC-M-20X, Newport, CA, USA), placed immediately under the sample holder, a mirror (RC-10Q620B, Newport, CA, USA) aligned with a zoom lens (VIDEO LENS- VZM 450, Edmund Optics, York, UK) that was connected to a CMOS camera (EO-1312C COLOR USB CAMERA, Edmund Optics, York, UK). Then, the micro-lensed polymeric probes were dipped in the sample under analysis. The laser light was sinusoidally modulated with a frequency of 1 kHz and a sampling rate of 100 kHz, and guided through the optical fiber to the sample, where it was focused by the microlens. By moving the relevant information to the 1 kHz band, this modulation allowed us to avoid undesired low frequency contributions to the signal, such as the 50 Hz local electrical grid component. The importance of this procedure was already demonstrated elsewhere [16,21,22,23]. The average optical power to which samples were exposed was 13 mW, an optimized optical power that minimized sample damage while maintaining the overall quality of sample analysis. The signal was acquired for 30 s with an average optical power of 13 mW, an optimized optical power that minimized sample damage while maintaining the overall quality of sample analysis. During acquisition, light was guided through the fiber to the photodetector and the signal digitalized with a DAQ (NI USA-6361, X Series DAQ, National Instruments, TX, USA) at a sampling rate of 10 kHz. The Multifunction I/O (Input/Output) device (NI USA-6361) controlled the optoelectronic components through a customized MATLAB Data Acquisition Toolbox script that simultaneously activated one Voltage-Output Port and one Voltage-Input Port. On the output port, it applied the base signal for the modulation of the laser, whereas the input port read the signal that reached the photodetector. After the acquisition was completed, the tip was moved to a different (x, y, z) position to be exposed to different nano-scale environments (characterized by distinct analytes concentration and microscopic refractive index distribution) within the same sample and the process was repeated until a total of 10 signals of 30 s were acquired. These repeated measurements were performed to guarantee that distinct microenvironments of the sample were considered in the final sample representation, enabling a biophysical mapping of the sample that maximized the identification of patterns that characterized each sample. Afterwards, the tip was removed and dipped in a bleach solution (20%) followed by distilled water, to avoid cross-contamination. This completed the sample analysis process.

### 2.5. Signal Processing

The acquired digital signal was processed using a custom-made python routine. First, it was phase matched by truncating the signal, so it started and ended on the 1 ms period minimum point. The minimum point was chosen as the moment at each period where the backscattered signal should have less variance [24]. The signal was then filtered using a second-order Butterworth high-pass filter with a cut-off frequency of 500 Hz to remove noisy, low-frequency components. The phase matching improved the consistency of the filtering process, as the signal limits could be filtered with much fewer artifacts using an even padding. Lastly, the signals were normalized using the z-score [25].

### 2.6. Feature Definition

After processing, each signal acquisition was split into three portions of 10 s and a set of 98 time- and frequency-related features (Appendix A; Extended methods) were calculated for each segment using a custom-built python script. Time-derived features comprised time-domain metrics and non-linear measures [26]. While time-domain metrics provided information on distinct statistical aspects of the signal, non-linear measures described the complexity and regularity of the signal. On the other hand, frequency-related features were subdivided into wavelet-derived, DCT-derived, and spectral [27,28]. Altogether, these features captured the behavior of the signal in the different frequency bands [29].

Patients’ demographics, clinical, and laboratory variables were encoded into thirteen distinct features (Appendix A), including age, number of comorbidities and presence of most frequent comorbidities associated with severe disease, and clinical information. Numeric variables were encoded using their numeric value (e.g., age). The number of kidney, cardiovascular, immunosuppressive, and respiratory diseases per patient obtained from their clinical records was also encoded as a numerical variable. Features related to the presence of specific comorbidities (diabetes, obesity) and IgG detection were calculated using one-hot encoding (0 = not present/detected; 1 = present/detected).

### 2.7. Model Development

Before developing the models for COVID-19 severity prediction, the dataset was randomly split into training and testing subsets, containing 80% and 20% of the data, respectively. This split was performed assuring that the data of each patient belonged uniquely to one subset. Thereafter, the training set was resampled to balance the two classes (i.e., severe and non-severe) and the features normalized using z-score. During model training, four different learning algorithms were tested: SVM, XGBoost, Random Forest, and Gradient Boosting Decision Trees. The SVM algorithm held the best generalization capacity, likely due to the low ratio between sample size and number of features used in this experiment. Three baseline SVM models were developed using (1) the features derived from the backscattered signal (Appendix A) comorbidities-based features and age information (Appendix A) comorbidities, age, and laboratory data available for each patient (Appendix A). Each model was independently trained using a fivefold cross-validation strategy and fine-tuned to find the ideal model hyperparameters (C, gamma, and kernel). The best models were selected as those that maximized AUROC across all folds. The performance of each selected model was evaluated on the test set; predicted classes from each model were compared with the ground-truth to determine the AUROC, accuracy, sensitivity, specificity, and precision measures. To develop the final stacking ensemble model [30], the individually optimized backscattered information (1) and comorbidities and age (2) models were used as base learners. A meta-model (Logistic Regression) was then fitted using the probability predictions made by both models as input; the combination of these inputs produced a new set of predictions. This strategy enabled the combination of the capabilities of the two base models and, consequently, an increased predictive power of the final model [30]. As previously, the performance of this model was evaluated based on the AUROC, accuracy, sensitivity, specificity, and precision on the test set. Finally, the contribution of each feature and its impact on the outcome of the model outcome was calculated using SHaply Additive explanations (SHAP) scores [31].

A simultaneous model was created to distinguish COVID-19 patients from patients presenting other infections with similar symptoms. Signal processing and feature calculation was performed as previously mentioned, however, some changes in the methods were applied as they boosted the performance of this model in the validation set. After feature normalization, the best 42 features with Fisher score above 0.025 were selected [32]. These features were then used to train a Random Forest model (from sci-kit learn python library) with 120 trees with a maximum level of 2. Each tree had a bootstrapping of samples, and six features were considered per split, to guarantee more variability between them and reduce the chance of overfitting. These hyperparameters were chosen using a grid search method where different models were evaluated in a validation dataset, separate from the test set. The performance of this model was also evaluated based on the AUROC, accuracy, sensitivity, specificity, and precision on the test set.

### 2.8. Statistical Analysis

Continuous variables (age, C-reactive peptide, leukocyte count, and lymphocyte count) were presented as median and interquartile range, while categorical variables (gender, comorbidities, and IgG detection) were shown as frequency and percentage. Differences between severe and non-severe groups were analyzed using Mann–Whitney U-test for continuous variables, and Chi-square test or Fisher’s exact test for categorical variables. Two-sided p-values were reported, and a significance level of 0.05 was considered for statistical analysis. The discrimination performance of the machine learning models was quantified by the AUROC; accuracy, sensitivity, specificity, and precision were also used as performance measures for the machine learning models. The confidence interval for AUROC was calculated using the Binomial exact confidence interval, as for the remaining metrics (accuracy, sensitivity, specificity, and precision) the Clopper–Pearson confidence interval was applied [33]. Comparison between AUROC of the stacking model with the baseline and SVM model including all clinical data was performed by DeLong statistical test [34]. The Spearman correlation was used in the correlation analysis. The performance measurements of the machine learning models were calculated in the testing dataset (an independent set from both the training and validation sets). Statistical analysis and representation was performed using Python programming language (pandas, numpy, sci-kit learn, and scipy.stats for analysis; matplotlib and seaborn for representation) and GraphPad Prism 8 v8.2.1 (GraphPad Software, San Diego, CA, USA).

## 3. Results

### 3.1. Study Design and Cohort Characteristics

A total of 88 patients admitted at Centro Hospitalar Universitário São João (CHUSJ) with confirmed COVID-19 disease were enrolled in this prospective study (Figure 1). COVID-19 positive diagnosis was confirmed by real-time reverse transcription-polymerase chain reaction (RT-PCR) assay of SARS-CoV-2 from nasopharyngeal swab specimens. Patients presented early/mild symptoms of disease at the moment of diagnosis, and hospital admission was decided considering the demographic, clinical, and laboratorial patient analysis performed during diagnosis. Serum samples analyzed by iLoF were collected at this moment, when the primary outcome of the study, i.e., patient admission to the intensive care unit (ICU), was still unknown. Patients who progressed to requiring intensive care (*n* = 38) during the follow-up period were included in the severe group, and patients who required hospitalization but not intensive care (*n* = 25), or who recovered at home (*n* = 25) were included in the non-severe group (*n* = 50). This progression information was considered to develop the prediction models of COVID-19 disease severity.

The clinical and demographic characteristics of severe and non-severe patients of the cohort are presented in Table 1. Compared with those in the non-severe group, there was a significantly higher prevalence of males (*p* = 0.0087), patients with cardiovascular disease (*p* = 0.0030), diabetes (*p* = 0.0091), kidney disease (*p* = 0.0157), and immunosuppression conditions (*p* = 0.0496) in the severe group. The reported number of comorbidities per patient was also significantly higher in patients who developed severe illness (*p* = 0.0051). No significant difference was found in age or other comorbidities, such as obesity and respiratory diseases, between the two groups. Severe patients had significantly higher levels of C-reactive peptide (*p* < 0.0001), leukocyte count (*p* = 0.0112), and were IgG positive to SARS-CoV-2 (*p* = 0.0006), following the observations previously reported [35,36,37].

### 3.2. Optical Fingerprinting of COVID-19 Serum Samples Provides Unique Information for Severity Prediction

We first investigated whether the features calculated from the backscattered signal of patient serum samples were predictive of COVID-19 disease severity. To achieve this, patient serum samples were exposed to a highly focused laser beam (λ = 976 nm), conducted by a micro-lensed single mode optical fiber (Appendix A). The backscattered signal resulting from the interaction between the irradiation light and micro/nanoparticles present in the sample, was collected through the same optical fiber and guided to a photodetector. Upon conversion to a digital signal, time- and frequency-domain features (Appendix A; Extended Methods) were obtained and used as input to a supervised machine learning model.

Before training the model to classify patient serum samples into severe or non-severe classes, the dataset was randomly split into training (80% of the data) and testing (20% of the data) subsets. The training set was resampled to balance the two classes (i.e., severe and non-severe) and the features normalized using z-score. Several learning algorithms were analyzed, including XGBoost, Random Forest, and Gradient Boosting Decision Trees; however, the support vector machine (SVM) algorithm held the best generalization capacity, likely due to the low ratio between sample size and number of features used in this experiment. The SVM model was trained and evaluated with fivefold cross-validation, and maximization of AUROC across all folds was the criterion followed to choose the best model. The fine-tuned model hyperparameters (C, gamma, and kernel) are indicated in Appendix A. In the testing set, the performance of the optical fingerprint in distinguishing severe and non-severe patients was assessed by both precision-recall (PR) and receiver operating characteristic (ROC) curves (Figure 2a,b). The area under the ROC curve (AUROC) was 0.63 (95% CI: 0.58–0.68) in the testing dataset, showing that the SVM model based on the optical fingerprinting data had a moderate prediction performance.

The contribution and effect of the optical fingerprinting features for the prediction of COVID-19 severity were measured by SHaply Additive explanations (SHAP) scores (Figure 2c). The SHAP scores estimated the relative importance of each optical feature in predicting severe COVID-19 disease. The “spectral roll-off” standard deviation (SD) was one of the most important optical features to predict disease severity. This feature estimates the variation in the energy of the frequency spectrum over time. Low values of spectral roll-off SD (represented in blue; Figure 2c), had a high contribution to predicting a severe outcome. Following an opposite trend, high values of singular value decomposition (SVD) entropy (represented in red; Figure 2c)—a measurement of the complexity of the backscattered signal—had high and positive importance (positive SHAP value) to describing a severe outcome. Similarly, high skewness values of the backscattered signal also had a positive contribution to classify patients in the severe group, however, the relative feature contribution was decreased when compared to SVD entropy.

To assess whether optical fingerprinting was independent from other variables collected at patient diagnosis, we analyzed the correlation between the model numeric output and patient laboratory variables. The model probabilistic output, taking values in the range 0 and 1, indicates the certainty with which a model outputs a prediction; the closer the output is from the extremes of the range, the more confident the model is on its classification. Of notice, the model numeric output was not strongly correlated with the available individual clinical data on inflammatory indexes, including IgG detection (*p* = 0.4917; Figure 2d), C-reactive protein levels (*r*^2^ = 0.02; *p* = 0.0004; Figure 2e), leukocyte (*r*^2^ = 0.21; *p* < 0.0001; Figure 2f) and lymphocyte counts (*r*^2^ = 0.15; *p* < 0.0001; Figure 2g). These results suggest that none of these individual clinical parameters routinely measured at diagnosis could inform on the specific biological conditions that were being captured by the optical fingerprint. Moreover, these findings indicate that the features extracted from the optical fingerprinting of serum patient samples may provide unique information for early severity risk assessment.

### 3.3. Stacking Ensemble Model Based on Optical Fingerprinting, Comorbidities and Age Information

We then evaluated to what extent patient information that can be easily collected at diagnosis, such as comorbidities data and age, would improve the performance of the optical fingerprint prediction model. To accomplish this, we created a stacking ensemble model combining the predictions of the optical fingerprinting SVM model and an additional comorbidities SVM model. This stacking model applies a logistic regression algorithm to learn how the predictions of the two baseline models can be combined to achieve a better performance [38]. The performance and main features of the stacking ensemble model are shown in Figure 3. In comparison to the baseline models, the stacking model displayed enhanced performance both regarding PR (Figure 3a) and ROC curves (AUROC = 0.80, 95% CI: 0.76–0.84; Figure 3b and Appendix A). These results represent a significant improvement of 0.06–0.17 in the discrimination of severe and non-severe patients (*p* < 0.0001, stacking vs optical fingerprinting model; *p* = 0.0030, stacking vs comorbidities/age model; Table 2). The two baseline models showed similar accuracy (~65%), however, the optical fingerprint model had slightly higher precision as compared to the comorbidities/age model (67% and 61%, respectively). The comorbidities/age baseline model had a sensitivity of 100%, i.e., all severe cases were predicted as severe, but demonstrated poor specificity (25%), suggesting that a large proportion of non-severe cases could be incorrectly predicted as severe. As for the optical fingerprint baseline model, both the sensitivity and the specificity were above 62%. Combining the probabilistic predictions of these models, the stacking model displayed a sensitivity and specificity above 85% and 72%, respectively. These results represent an improvement of approximately 18% in sensitivity and 11% in specificity, as compared to the optical fingerprinting model; and an improvement of 47% in specificity, in comparison to the comorbidities/age model. The feature importance analysis of the stacking model (Figure 3c) shows that comorbidities related features, namely kidney, cardiovascular diseases, and immunosuppression conditions, had a high impact on the prediction of severe cases, which may contribute to the increased sensitivity (Figure 3a).

To further assess the clinical utility of the optical fingerprint, we developed an additional SVM model including all the clinical and laboratory patient data available at hospital admission, i.e., IgG detection, C-reactive peptide levels, leukocyte and lymphocytes counts, as well as patient comorbidities and age (Table 2; Appendix A). Notably, the stacking model, including the optical fingerprinting and comorbidities/age information, performed significantly better than the model encompassing patient clinical and laboratory data, with an increase of 0.05 in the discrimination of the two classes (*p* = 0.0322; Table 2)

Altogether, these results suggest that a single optical fingerprinting analysis of patient serum samples may provide a better predictive discrimination of severe and non-severe patients as compared to the combination of several routine blood analyses.

The optical fingerprinting collected at the moment of patient diagnosis provided a holistic and agnostic analysis of patient-derived blood samples that might potentially be used to address distinct clinical or biological questions. We further investigated whether the holistic and agnostic information provided by the optical fingerprinting could be used to distinguish COVID-19 patients from patients presenting with similar symptoms. To accomplish this, we performed a supplementary analysis of the features extracted from the optical fingerprinting of 38 patients (Appendix A), including patients diagnosed with COVID-19 only (COVID-19 group; *n* = 19; patients sampled from the original cohort) and COVID-19 negative patients with other respiratory infections (non-COVID-19 group; *n* = 19). To avoid overfitting due to the reduced sample size, a random forest algorithm was applied to distinguish the two classes using the optical fingerprinting-derived information only. Training and testing were performed as previously mentioned. The optimized model (Appendix A), maximizing the AUROC across the validation folds, achieved an AUROC of 0.88 (95% CI: 0.84–0.92; Appendix A), and all performance measurements were above 75% (Appendix A). There was no correlation between the model prediction and patient age (*r*^2^ = 0.22; *p* < 0.0001; Appendix A) or gender (*p* = 0.1649; Appendix A).

These preliminary findings suggested that the same optical fingerprinting data, extracted from the unique backscattered signal produced by patient serum samples, can be used to feed simultaneous machine learning models addressing multiple clinical challenges in patient diagnosis and prognosis.

Overall, this work demonstrated that optical fingerprinting of patient serum samples collected at the moment of diagnosis, together with patient age and comorbidities information, can be integrated to provide an accurate severity risk assessment tool for COVID-19 disease (Figure 4). This severity forecast information may further enable patient triage and support clinical decision making regarding patient surveillance and treatment timing. Hence, this technology may provide a unique opportunity to develop a holistic and agnostic platform to manage the current COVID-19 pandemic, as well as to quickly respond to potential infectious disease outbreaks in the future.

## 4. Discussion

In this study, we combined photonics and machine learning algorithms to develop an innovative and holistic method to predict disease severity of COVID-19 patients at the moment of diagnosis. Our findings suggest that optical fingerprinting of patient serum samples, together with information on patient comorbidities and age, provide a robust and reliable tool to support the triage of COVID-19 patients that will require ICU admission.

The urgent need to provide an efficient diagnosis and prognosis for COVID-19 patients has guided the development of a high number of scoring systems and artificial intelligence-based prediction models. Most prognosis models apply logistic regression or machine learning methods to patient demographics, laboratory, and clinical data, collected at hospital admission [37,39,40,41,42]. Although these methods achieve high discrimination levels, the lack of standardized procedures to register patient information and perform routine laboratory analysis hampers the generalization of these models to external healthcare systems [43]. Moreover, other blood-based biomarkers that are not measured in routine laboratory analyses, may also inform on COVID-19 severity, namely viral RNA load, mitochondrial DNA levels, and red blood cell distribution width [44,45,46]. The optical fingerprinting data herein studied represents the sum of each structure’s light scattered signal contribution. When the laser beam is incident on a sample, small structures scatter the incident light and are exposed to gradient forces induced by the optical harmonic potential, generated by the electromagnetic field that is propagated by the micro-lensed fiber tip. The gradient component of the optical force exerts a restoring force that disturbs the intrinsic Brownian motion of the particles, deviating them from their baseline behavior [47]. Due to the small size of the target particles (smaller than the wavelength of the irradiation light, i.e., 976 nm), this harmonic trapping potential is not sufficient to cause optical particle trapping; however, the resulting Brownian fluctuations, intrinsically related to the optical and biophysical properties of the particles (e.g., size, shape, microscopic refractive index, biochemical composition), will contribute to a specific light scatter patterning [34,47,48]. Hence, as each serum sample is comprised by a unique combination of particles, the resulting backscattered pattern is also a unique representation of the sample. Once the backscattered pattern is collected as a digital signal, supervised machine learning methods can be applied to the features of the signal to identify patterns that may be predictive of COVID-19 severity, enabling sample classification. Hence, the main advantage of this agnostic analysis is the balanced and simultaneous contribution of multiple biomarkers. Nonetheless, this also means that many unrelated COVID-19 biomarkers will contribute to the signal, which highlights the need for a sufficient sample number to reduce the impact of non-specific sample variability. As such, the performance of the prediction model can be expected to improve as the number of samples involved is increased. We anticipate that by increasing sample size, it will be possible to fine-tune feature importance and reduce the overall number of features considered by the optical fingerprinting model to forecast COVID-19 severity. Even so, our current approach demonstrated a high predictive level in a relatively small cohort, which disclosed the great potential of this technology.

Alongside blood-based parameters, age and comorbidities have been shown to improve the performance of severity prediction models [43]. Thus, we developed a stacking ensemble model combining the digital information from the optical fingerprinting and the reported patient comorbidities and age. We considered a restricted set of comorbidities that have been widely associated with poor prognosis and disease severity of COVID-19 patients, namely immunocompromised conditions, diabetes mellitus, obesity, chronic kidney, cardiovascular, and respiratory diseases [5,6,7,8]. Patient information on comorbidities and age can be easily accessed through electronic health records or through a quick questionnaire at the moment of diagnosis. By combining the best predictions of each individual model, the final stacking ensemble model outperformed the comparative model based on all patient clinical and laboratory information available at diagnosis. Hence, the iLoF platform provides an accurate COVID-19 severity forecasting tool without requiring further laboratory or clinical assessments.

Lastly, we provided evidence that the iLoF agnostic and holistic analysis of COVID-19 patient serum may have a broader clinical utility. When comparing the optical fingerprint of patients with similar respiratory infection symptoms, our tool demonstrated a good performance in distinguishing COVID-19 patients from patients with other causes of infection. These results will be further explored in a larger cohort, including higher representation of common causes of respiratory infection to develop a pre-screening model that might support diagnosis of respiratory diseases. Such a tool may have a strong impact in the early detection of and response to future outbreaks of infectious diseases.

The iLoF technology can be easily adapted to the temporal and spatial dynamics of worldwide healthcare systems. This low-cost and minimally invasive tool does not require specialized training and is easy to operate, rendering accessibility to any healthcare unit. Moreover, patient serum samples can be analyzed in approximately 30 s using a portable device, which enables a fast-screening approach. The model developed in this study will be automatically improved as more data is acquired in different clinical settings, which may improve generalization to other healthcare systems.

Nevertheless, the present study also encompasses some limitations. The model was developed from a relatively small cohort and a single medical center, which limits the analysis on the generalization and validity in different clinical settings. To overcome this, the method is currently being tested in three independent validation cohorts. In the study design, the primary outcome was admission to ICU, however, the criteria for ICU admission may vary across medical centers and may be altered according to occupancy. Of notice, the COVID-19 severity model developed in this study can be adjusted automatically or with minimal effort to such clinical situations, provided that a sufficient sample size is acquired in these scenarios. The observational nature of this study precludes the inference of causality, and further studies are required to understand the biological factors that define the optical fingerprint of COVID-19 patients presenting distinct disease severity.

## 5. Conclusions

To conclude, the method herein described demonstrates that patient blood optical fingerprinting, combined with information on age and comorbidities, predicts COVID-19 disease severity with a 0.80 AUROC and 80% accuracy. The low-cost, minimally invasive, and rapid pipeline required to collect the optical fingerprint of COVID-19 patients renders the use of this technology as a front-line screening platform to support clinical decision making. Furthermore, the innovative holistic and agnostic analysis might be modelled to provide simultaneous information to assist COVID-19 patient diagnosis and prognosis.

## Figures and Tables

**Figure 1 diagnostics-11-01309-f001:**
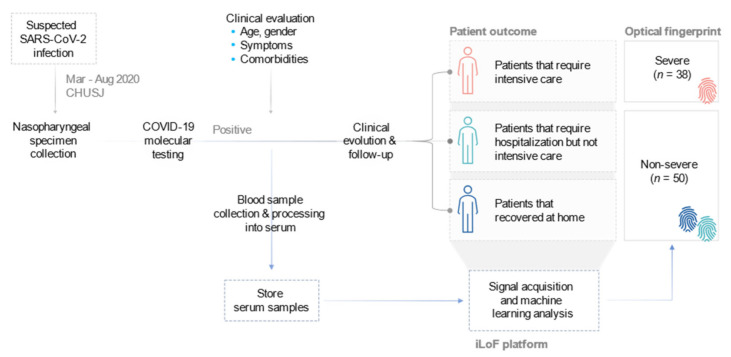
Study design. Patients admitted at the CHUSJ with positive COVID-19 diagnosis by RT-PCR molecular testing were enrolled in the study. Peripheral blood was collected, processed into serum, and analyzed using the iLoF platform. The primary outcome was measured up to four weeks after diagnosis, with 38 patients developing severe disease that required ICU admission, and the remaining 50 patients presenting mild to light symptoms, thus not requiring intensive care.

**Figure 2 diagnostics-11-01309-f002:**
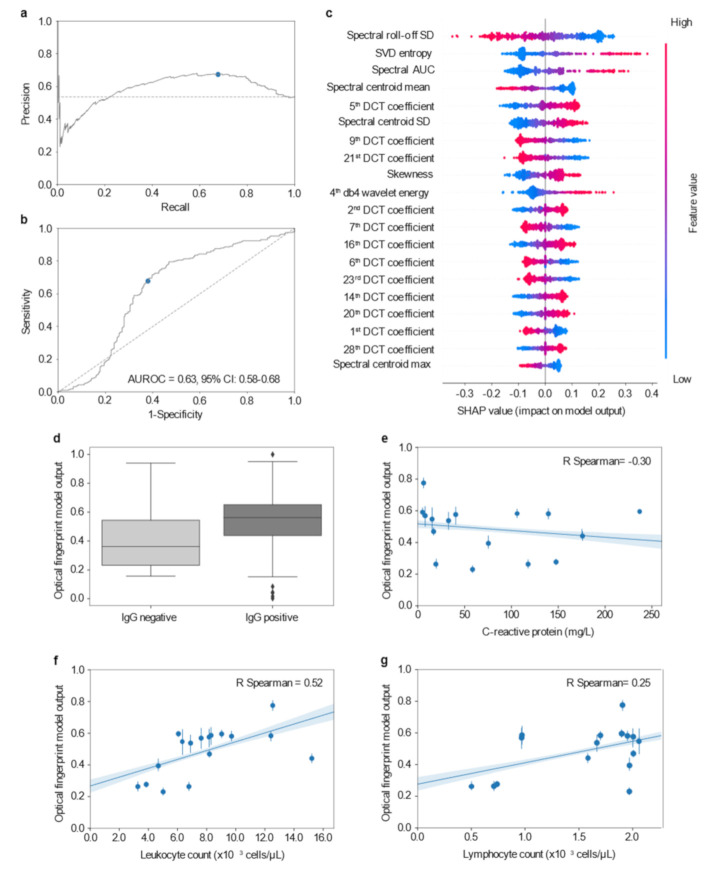
Establishment of the optical fingerprint of COVID-19 severe patients. (**a**) PR plot showing the positive predictive value (precision) against the sensitivity (recall) of the optical fingerprint model. (**b**) ROC curve showing the trade-off between sensitivity and specificity using the optical fingerprint model. The diagonal dashed line represents a model with no discrimination. The AUROC with its 95% confidence interval is shown in the plot. Both PR and ROC curves were obtained from the test dataset; blue dots indicate the optimal threshold, while dashed lines represent a random prediction model. (**c**) Summary plot of the SHAP values of the top 20 optical features distinguishing severe and non-severe patients. Features are ranked (top to bottom) by their overall importance in contributing to the final prediction. The color of each point represents the value of the predictor, with the higher values corresponding to red and the lower values to blue. The distribution along the X-axis indicates the effect that a feature had on the prediction of that specific case. The positive range indicates that the feature contributed to increase the risk of severe illness prediction. (**d**–**g**) Association and correlation by linear regression between the optical fingerprinting prediction and clinical inflammatory parameters including detection of IgG (**d**), levels of C-reactive peptide (**e**), Leukocyte (**f**), and lymphocyte (**g**) counts. (**e**–**g**) Pearson correlation coefficients were calculated for each linear regression and are shown in the respective plots. AUROC: area under the receiver operating characteristic curve; SD: standard deviation; SVD: singular value decomposition; AUC: area under the curve; DCT: discrete cosine transform.

**Figure 3 diagnostics-11-01309-f003:**
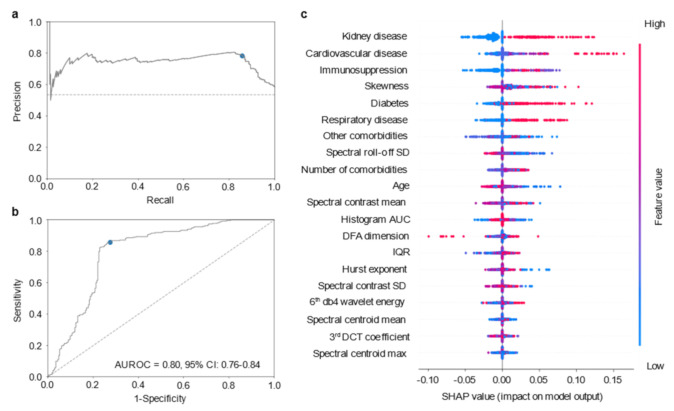
Optical fingerprint-based model to forecast COVID-19 severity. (**a**) PR plot showing the positive predictive value (precision) against the sensitivity (recall) of the optical fingerprint/comorbidities/age stack model. (**b**) ROC curve showing the trade-off between sensitivity and specificity using the optical fingerprint/comorbidities/age stack model. The AUROC with its 95% confidence interval is shown in the plot. Both PR and ROC curves were obtained from the test dataset; blue dots indicate the optimal threshold, while dashed lines represent a random prediction model. (**c**) Summary plot of the SHAP values of the top 20 most important features to distinguish severe and non-severe COVID-19 patients using the stack model of optical fingerprint and comorbidities/age. AUROC: area under the receiver operating characteristic curve; SD: standard deviation; AUC: area under the curve; IQR: Interquartile range; DCT: discrete cosine transform.

**Figure 4 diagnostics-11-01309-f004:**
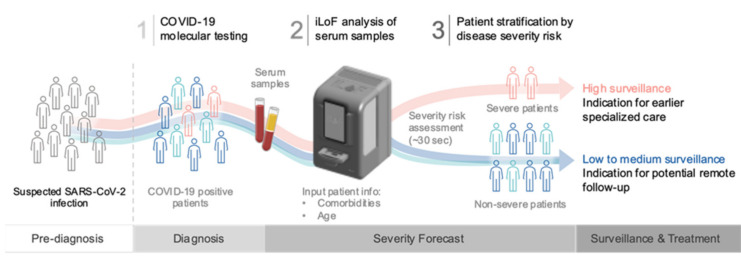
Impact of the optical fingerprinting, comorbidities and age stacking model in the clinical trajectory of COVID-19 patients. Optical fingerprinting of patient serum samples, together with the information on patient comorbidities and age, provide a robust and reliable severity risk assessment tool to support the stratification of COVID-19 patients immediately after diagnosis. This information might be of utmost importance to define patient surveillance, manage the timing for treatment initiation, and assist resource allocation.

**Table 1 diagnostics-11-01309-t001:** Demographics and clinical characteristics between the severe and non-severe patients.

Variable	Severe Group *(n = 38)*		Non-Severe Group *(n = 50)*		*p* Value
	Patients	Missing Data	Patients	Missing Data	
***Age (y), median (IQR)***	71 (61–77)		64 (46–83)		0.2870
Gender (Male)	28 (73.7%)		22 (44.0%)		**0.0087**
Comorbidities, *N (%)*		1 (2.6%)		5 (10.0%)	
Kidney disease	10 (27.0%)		3 (6.7%)		**0.0157**
Cardiovascular disease	29 (78.4%)		20 (44.4%)		**0.0030**
Immunosuppressed	11 (29.7%)		5 (11.1%)		**0.0496**
Diabetes	18 (48.6%)		9 (20.0%)		**0.0091**
Respiratory disease	9 (24.3%)		4 (8.9%)		0.0724
Obesity	7 (18.9%)		2 (4.4%)		0.0714
Others	19 (51.4%)		24 (53.3%)		>0.9999
Comorbidities per patient, *N* (%)					**0.0051**
0	3 (7.9%)		13 (26.0%)		
1 or 2	9 (23.7%)		15 (30.0%)		
3 or 4	11 (28.9%		13 (26.0%)		
≥5	14 (36.8%)		4 (8.0%)		
C-reactive peptide (mg/L), *median (IQR)*	117.0 (46.1–190.6)		21.1 (6.5–69.8)		**<0.0001**
Leukocyte count (×10^3^ cells/uL), *median (IQR)*	7.6 (6.0–10.9)		5.6 (4.5–8.3)		**0.0112**
Lymphocyte count (×10^3^ cells/uL), *median (IQR)*	1.1 (0.6–1.7)		1.4 (0.8–1.8)	1 (2.0%)	0.1645
IgG (Positive), *N* (%)	28 (75.7%)	1 (2.6%)	19 (38.0%)		**0.0006**

Data presented as median (IQR) or *n* (%). Differences between groups were analyzed using the Mann–Whitney U-test for continuous variables and the Chi-square test or Fisher’s exact test for categorical variables. Two-sided p-values are reported. IQR: interquartile range.

**Table 2 diagnostics-11-01309-t002:** Statistical performance measures of the baseline and stacking models predicting COVID-19 severity.

Model	AUC	Accuracy	Sensitivity	Specificity	Precision
	(95% CI)	(95% CI)	(95% CI)	(95% CI)	(95% CI)
Optical fingerprint, comorbidities, age	0.80	79.6%	85.7%	72.6%	78.3%
	(0.76–0.84)	(75.8–83.0)	(81.0-89.6)	(66.4–78.2)	(73.2–82.8)
Optical fingerprint	0.63 ****	65.1%	67.8%	62.0%	67.3%
	(0.58–0.68)	(60.8–69.2)	(61.9–73.3)	(55.5–68.2)	(61.4–72.8)
Comorbidities, age	0.74 **	65.3%	100.0%	25.3%	60.7%
	(0.70–0.78)	(61.0-69.4)	(98.9–100.0)	(11.4–31.4)	(56.0–65.2)
All clinical data(Comorbidities, age, laboratory parameters)	0.75 *	76.5%	78.0%	74.7%	78.0%
	(0.68–0.76)	(72.5–80.1)	(72.6–82.8)	(68.2–80.1)	(72.6–82.8)

Performance measures calculated for the testing dataset. Binomial (AUROC) or Clopper–Pearson (accuracy, sensitivity, specificity, and precision) confidence intervals were calculated in the testing dataset. The DeLong statistical test was performed to compare the AUROC of the stacking model (optical fingerprint, comorbidities, age) with each of the baseline models and the clinical data SVM model; * *p* < 0.05, ** *p* < 0.01, **** *p* < 0.0001. AUROC: area under the receiver operating characteristic curve; CI: confidence interval.

## Data Availability

Data are contained within the article and/or Appendix A. Additional data presented in this study are available on request from the corresponding author.

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
