# Peer review of "Forecasting COVID-19 Severity by Intelligent Optical Fingerprinting of Blood Samples"

_diagnostics, 2021, doi:10.3390/diagnostics11081309_

Round 1

Reviewer 1 Report

Comment 1 (Section 2.4) 

Authors state "The average optical power to which the samples were exposed was 13 mW". Please, it is possible to provide further details about?

Comment 2 (Section 2.4)

Authors state "The signal was then acquired for 30 seconds using Matlab’s Data Acquisition Toolbox in a custom-made script".  Please, it is possible to provide further details about?

Comment 3 (Section 2.4)

Authors state "After the acquisition was completed, the tip was displaced to a different position ...".  Please, it is possible to provide further details about?

Comment 4 (Section 2.6)

How are encoded demographic and clinical features shown in Talbe S2?

Comment  5 (Section 2.7)

'SHAP' acronym is used without have been introduced.

Reviewer 2 Report

Dear Authors,

Overall, congratulations to the authors on their findings.

The usefulness of the research and results cannot be questioned.

I cannot comment on the content of the manuscript from a medical point of view, but as an engineer, focusing on engineering and statistical analysis areas.

The diagnostic method presented is based on an innovative approach. The manuscript includes the operation of the entire decision support platform from the first sampling to the evaluation of the results in a completely understandable and precise manner.

The chapters of the manuscript are logically structured, the structure is entirely appropriate for presenting the process and results of scientific research.

The sources referenced in the manuscript match the content, current, up-to-date sources.

The goodness of the defined model is supported by the results of the statistical analysis.

The documentation uploaded as supplementary material contains a lot of useful information about the process related to the complete evaluation, its mathematical and digital signal processing background.

Congratulations to the authors, I wish them much success in their further research in a similar field.

Best regards

Author Response

The authors are deeply grateful to the Reviewer for the encouraging and outstanding comments and enthusiasm regarding the work presented in this manuscript and regarding our technological approach. We genuinely appreciate your vision of our work, and we are happy that the present manuscript fulfilled the expectations and high standards of the Reviewer. We hope that our work will gather the same enthusiasm from future readers. Many thanks for having accepted reviewing our manuscript.
Best wishes

Round 2

Reviewer 1 Report

Not all comments raised in the previous review round were approached. The authors have only responded to the reviewer, without integrating the paper.

In particular, the following two comments weren't addressed.

Comment 1 (Section 2.4): Authors state "The signal was then acquired for 30 seconds using Matlab’s Data Acquisition Toolbox in a custom-made script". Please, it is possible to provide further details about?

Comment 2 (Section 2.4): Authors state "After the acquisition was completed, the tip was displaced to a different position ...". Please, it is possible to provide further details about?

Author Response

The authors appreciated the observation and included further modifications in the manuscript to address the comments 1 and 2 from the Reviewer.

Specifically, we have included the following modification in the manuscript section 2.4, which are highlight in the revised version of the manuscript that was now resubmitted.

Regarding comment 1:

Page 4, §163

“The signal was acquired for 30 seconds with an average optical power of 13 mW, an optimized optical power that minimized sample damage while maintaining the overall quality of sample analysis.  During acquisition, light was guided through the fiber to the photodetector and the signal digitalized with a DAQ (NI USA-6361, X Series DAQ, National Instruments, TX, USA) at a sampling rate of 10kHz. The Multifunction I/O (In-put/Output) device (NI USA-6361) controls the optoelectronic components through a customized MATLAB Data Acquisition Toolbox script which simultaneously activates one Voltage-Output Port and one Voltage-Input Port. On the output port, it applies the base signal for the modulation of the laser, whereas the input port reads the signal which reaches the photodetector.”

Regarding comment 2:

Page 4, §172

“After the acquisition was completed, the tip was moved to a different (x, y, z) position to be exposed to different nano-scale environments (characterized by distinct analytes concentration and microscopic refractive index distribution) within the same sample and the process was repeated until a total of 10 signals of 30 seconds were acquired.  These repeated measurements were performed to guarantee that distinct microenvironments of the sample are considered in the final sample representation, enabling a biophysical mapping of the sample that maximize the identification of patterns that characterize each sample.”

Round 3

Reviewer 1 Report

All comments were approached.